# Targeting Cx43 to Reduce the Severity of Pressure Ulcer Progression

**DOI:** 10.3390/cells12242856

**Published:** 2023-12-18

**Authors:** Milton Sheng Yi Kwek, Moogaambikai Thangaveloo, Leigh E. Madden, Anthony R. J. Phillips, David L. Becker

**Affiliations:** 1Lee Kong Chian School of Medicine, Nanyang Technological University Singapore, Clinical Sciences Building, 11, Mandalay Road, Singapore 308232, Singaporemoog0004@e.ntu.edu.sg (M.T.); lmadden@ntu.edu.sg (L.E.M.); 2Skin Research Institute Singapore, Clinical Sciences Building, 11, Mandalay Road, Singapore 308232, Singapore; 3School of Biological Sciences, Auckland University, Auckland 1010, New Zealand; a.phillips@auckland.ac.nz

**Keywords:** gap junction, Cx43, pressure ulcer, inflammation, necroptosis

## Abstract

In the skin, repeated incidents of ischemia followed by reperfusion can result in the breakdown of the skin and the formation of a pressure ulcer. Here we gently applied paired magnets to the backs of mice to cause ischemia for 1.5 h and then removed them to allow reperfusion. The sterile inflammatory response generated within 4 h causes a stage 1 pressure ulcer with an elevation of the gap junction protein Cx43 in the epidermis. If this process is repeated the insult will result in a more severe stage 2 pressure ulcer with a breakdown of the epidermis 2–3 days later. After a single pinch, the elevation of Cx43 in the epidermis is associated with the inflammatory response with an increased number of neutrophils, HMGB1 (marker of necrosis) and RIP3 (responsible for necroptosis). Delivering Cx43 specific antisense oligonucleotides sub-dermally after a single insult, was able to significantly reduce the elevation of epidermal Cx43 protein expression and reduce the number of neutrophils and prevent the elevation of HMGB1 and RIP3. In a double pinch model, the Cx43 antisense treatment was able to reduce the level of inflammation, necroptosis, and the extent of tissue damage and progression to an open wound. This approach may be useful in reducing the progression of stage 1 pressure ulcers to stage 2.

## 1. Introduction

Pressure ulcers (PU), also known as decubitus ulcers or bed sores, are the result of localised damage to the skin and underlying tissues following ischemia and reperfusion (IR) [1]. They have four stages with increasing severity [1]. Stage 1 involves a closed wound with non-blanchable intact skin [2]. This is largely due to an ischaemia reperfusion injury caused by prolonged pressure followed by reperfusion of blood on release [3,4]. Stage 2 is the breakdown of the epidermis and dermis. Stage 3 is full thickness skin loss involving damage to the dermis but not through underlying fascia. Stage 4 is full thickness with extensive damage to the muscle layer, fascia or exposed bone [2]. In the United States of America (USA), PU prevalence was reported to be 13.5% and 12.3% in 2008 and 2009, respectively [5]. In the USA alone, each patient incurs a cost of $500 to $70,000 USD, and the healthcare system cost is estimated to be $11 billion USD annually [6]. Development of a PU can be prevented by patient repositioning, nutritional intervention, special support, extra skin care and adopting risk assessment tools [7], but still they persist [8]. Numerous pharmaceutical approaches, aimed at preventing pressure ulcer progression, have been tested but have not proved to be effective [9,10].

When pressure is relieved the return of blood causes a reperfusion injury [11,12]. IR injury is viewed as the major contributing factor in the formation of PU [11] through the formation of reactive oxygen species and the resultant sterile inflammatory response [13]. Due to ischaemia, activation of Na^+^/Ca^2+^ ion exchangers results in increased intracellular Ca^2+^ which can activate death pathways such as apoptosis and necroptosis [14,15,16]. Necroptosis, is a cell death pathway that mimics apoptosis and necrosis and is involved in I/R injuries in organs such as brain [17,18], heart [19] and kidney [20]. Necroptosis is regulated by receptor interacting protein kinase 3 (RIP3/RIPK3) [21]. Upon activation, RIP3 is liberated from the RIP1/RIP3 complex and recruits MLKL to the cell membrane to form pores. Cell constituents then leak into the extracellular space causing the cells to die [22]. Necroptosis also triggers a robust inflammatory response that includes the release of high mobility group box-1 (HMGB1) [23,24]. HMGB1 is a damage associated molecular pattern (DAMP) that is involved in I/R injury in many organs such as kidney [25], liver [26], brain [27], heart [28] and the eye [29]. Upregulation of HMGB1 exacerbates I/R injury by stimulating the proinflammatory response [25,26,27,28,29,30]. In myocardial I/R injury, HMGB1 is significantly elevated as early as 30 min after ischaemia and peaks at 6 h post ischaemia [31] compared to 8 to 10 h after TNF stimulation in pituicytes [32] or 16 h after LPS administration in mice [33]. 

Gap junctions are intercellular channels that allow the passage of molecules of up to 1 kDa between cells [34]. In vertebrates, these channels are composed of the connexin family of proteins [35]. Six connexins form a connexon or hemichannel that docks to a connexon in an adjacent cell to form a gap junction [36]. Hemichannels are usually closed, but cells undergoing ischaemia reperfusion or oxidative stress can open the hemichannels releasing metabolites and secondary messengers such as adenosine triphosphate (ATP), nicotinamide dinucleotide (NAD), Ca^2+^, glutamate and inositol 1,4,5-triphosphate (IP3) [37,38,39,40]. Several types of connexins are expressed in the different layers of the epidermis and the dermis. Connexins found in the skin of humans and mice include Cx26, Cx30, Cx30.3, Cx31, Cx31.1, Cx37 and Cx43 [41,42]. Cx43 is the most highly expressed in the epidermis and dermis [42]. During skin repair, Cx43 expression changes dynamically during wound healing, decreasing in wound edge keratinocytes and fibroblasts to allow them to become migratory [43]. In human chronic wounds such diabetic foot ulcers, Cx43 expression is highly upregulated in the epidermis and dermis at the wound edge and migration fails [44]. In diabetic rat wounds Cx43 protein levels are elevated in keratinocytes at the wound edge and this underlies the delay in the wound healing process [45]. 

Following an ischemic stroke Cx43 expression is upregulated in astrocytes after reperfusion [46,47,48]. Gap junctions remain open during ischaemia and cells remain coupled during the period of cell death [49]. The gap junctions can spread death signals from dying cells to healthy neighbours in a “bystander effect” [50,51]. The inhibition of gap junctional communication can arrest neuronal death following a stroke [48,52]. In heart I/R models, blocking gap junction communication with a mimetic peptide (Gap19) reduced swelling in mouse cardiomyocytes, improved cardiomyocyte survival and reduced infarct size [53]. 

In this study, we show that the topical application of Cx43 antisense can reduce PU progression in mice by reducing I/R induced upregulation of epidermal Cx43. This also reduces necroptosis and the inflammatory response and mitigates the progression of experimental PU injury. 

## 2. Materials and Methods

### 2.1. Mice

All animal procedures were performed in accordance with Institutional Animal Care and Use Committee (IACUC) protocol numbers 140,928 and A0367. Male ICR-1 mice (Invivos, SG) 7–12 weeks old were anaesthetised with 4% isoflurane and 2 L/min oxygen and subcutaneous buprenorphrine (2 mg/kg) was administered for analgesia. The mouse backs were shaved, and a depilatory cream (Nair) was applied for 2 min, before removal with warm moist gauze. 

### 2.2. Mouse Magnet Double Pinch Ischaemia Reperfusion Model 

The mouse back skin was tented, and magnets (Gold plated neodymium; F656S, 12 mm diameter × 2 mm, 2000 Gauss, N42; Magnet Expert Therapy Magnets, UK) were gently applied either side of the skin, leaving a 1 cm skin bridge [54]. The location of the placement was identified with a marker pen. After the ischaemic period of 1.5 h, the magnets were gently removed to allow blood to return and cause a reperfusion injury. In some instances, after a period of 4 h of reperfusion the magnets were reapplied to the exact location for another 1.5 h of ischaemia and then removed for reperfusion, creating a double pinch ischaemia reperfusion model [54]. A 30% *w*/*v* Pluronic^®^ F-127 (Sigma, St. Louis, MO, USA) was made up in sterile Milli-Q^®^ water as vehicle control (VC). Cx43asODN 5′-GTAATTGCGGCAGGAGGAATTGTTTCTGTC-3′ (Sigma) was diluted in the Pluronic gel to achieve a final working concentration of 300 μM. An aliquot of 100 μL of the Cx43asODN gel, or vehicle control (VC) or sense control 5′-GACAGAAACAATTCCTCCTGCCGCAATTAC in 30% pluronic gel, was injected subcutaneously under the insult. 

### 2.3. Scoring 

Macroscopic images of the insulted areas were taken every day and adjusted for size prior to comparison. A severity score from 1 to 7, incorporating both size and redness (7 being the most severe), was devised to determine the severity of the injury. As the injuries healed, a guide was provided for scorers to avoid any misinterpretation of the extent of the injuries. All image sets were coded, and the scorer was blinded to the treatment groups. Three independent scorers then ranked the pictures in order of severity. 

### 2.4. Tissue Harvesting and Processing 

Animals were euthanized with CO_2_, and insulted skin biopsies were harvested along with normal skin as control. Samples were bisected, and one half fixed in 4% paraformaldehyde (PFA) for paraffin wax processing and the other half stored in RNALATER^TM^ for mRNA analysis. Tissues fixed in 4% PFA were transferred into 70% ethanol and processed using HistoCore Pearl (Leica, Wetzlar, Germany) through an ethanol concentration gradient followed by xylene (3 × 45 min, 45 °C) and paraffin (3 × 45 min, 62 °C) and embedded into paraffin blocks. Tissue was sectioned at 5 μm using a microtome (Leica RM2245, Germany). Slides were stained with haematoxylin and eosin (H&E) (Sigma, USA) using a Leica Autostainer XL (Leica, Germany). Slides were dewaxed in Clearene^TM^ (Leica Biosystems, Nußloch, Germany) and transferred through a series of ethanol and into water. Then, they were stained with haematoxylin and differentiated in 0.3% acid alcohol, washed in water, and transferred into Scott’s tap water before 70% ethanol and then Eosin. Slides were dehydrated and coverslip applied with Organo Limonene mounting media (Sigma, USA).

### 2.5. Immunofluorescence Staining

Slides were dewaxed and transferred through a graded series of ethanol and into water. Antigen retrieval with 0.01 M citrate buffer (10 mM Sodium Citrate, 0.05% Tween20, pH 6) at 60 °C overnight. Slides were washed with 1× PBS and permeabilised with 0.2% Triton X-100 in 1× PBS. Slides were blocked for 1 h in 1% bovine serum albumin (BSA) in PBS. They were stained with the respective primary antibodies in 1% BSA: (Rabbit anti-Connexin43, 1:1000 (C6219, Sigma), Rat anti-mouse Ly-6G, 1:500 (127602, Biolegend, San Diego, CA, USA), Rabbit anti-HMGB1, 1:500 (ab79823, Abcam, Cambridge, UK), Rabbit anti-RIP3, 1:500 (ab56164, Abcam), overnight at 4 °C. Slides were washed with PBS and incubated for an hour with respective secondary antibodies and counterstained with DAPI (1:10,000) and mounted on coverslips using Citifluor^TM^ (Citifluor Ltd., London, UK).

### 2.6. Confocal Imaging 

All confocal images were taken with a Leica SP8 confocal microscope (Leica, Germany) using a 40× objective (1.3NA, Leica). Three fields of view of 3 μm stack made up of 5 single planes with a max projection. Parameters of laser power gain and offset were kept constant within a group to allow for comparison. To quantify Cx43 or other signals, a threshold was set to only include positive signal and the image converted into a binary image and analysed by including all particles sized 1 upwards. To differentiate other structures from epidermis, a region of interest (ROI) was drawn manually to exclude the epidermis. The ROI was then analysed. Histological images and montages were taken using a Zeiss AxioscanZ1 slide scanner (Zeiss) using a 20× objective. 

### 2.7. Quantitative PCR

Total RNA was extracted from tissues stored in RNALater using RNeasy Fibrous Tissue Mini Kit (74704, Qiagen, Hilden, Germany). No more than 30 mg of tissue was used for RNA extraction. Subsequently, the cDNA from purified RNA was generated using SuperScript VILO cDNA synthesis kit (11754-250, Invitrogen, Waltham, MA, USA) recommended protocol. Validated PCR sequences of TNFα, IL6, MMP9 and GAPDH were obtained from Primerbank and purchased from IDT. The transcript levels of the genes of interest were measured by quantitative PCR according to manufacturer’s recommendation using Express STBR GreenER Kit, 11780-200 (Invitrogen, USA) and measured using StepOnePlus^TM^ Real-Time PCR Thermal Cycling Block (Applied Biosystems, Waltham, MA, USA). Deviations between technical replicates of >0.7 were excluded following value guides programmed into the qPCR machine (Applied Biosystems). Technical values were averaged and fold gene expression was the calculated by using formula 2^−ΔΔCt^. First, gene of interest Ct (cycle threshold) was normalised to their house keeping gene Ct value by taking the difference between them (ΔCt). The difference between an experimental sample and a control (treated vs. untreated) is (ΔΔCt). To find the expression fold change, values were brought up to the 2^−ΔΔCt^ and plotted for statistical analysis.

### 2.8. Statistics

All statistics were performed on the GraphPad Prism^®^ 6 software. When comparing between two independent, non-normalised groups, the Mann–Whitney U-test was used. When comparing two independent normalised groups, Wilcoxon signed-rank test was used. To compare between more than two independent groups, the non-parametric Kruskal–Wallis test followed by Dunn’s post hoc test was used. To compare between groups with 2 variables (score and time), a two-way ANOVA followed by Bonferroni’s multiple comparisons test was used. Data that was converted for Kaplan–Meier analysis was analysed using Log-rank (Mantel–Cox) test. A *p*-value of <0.05 was considered significant. All error bars denoted the standard deviation of the mean with the exception of 2-way ANOVA, which is denoted as standard error of mean.

## 3. Results

### 3.1. Double Cycle Pinch Model Results in a Stage 2 Pressure Ulcer

For the testing of treatments aimed at preventing the progression of a Stage 1 pressure ulcer to a Stage 2 pressure ulcer, a model capable of progression to an open wound was required. Using cycles of 1.5 h(I)/4 h(R)/1.5 h(I) resulted in an open wound on day 3. After opening, the wounds began to reduce in size and heal, which was completed by day 11.

### 3.2. Histological Analysis of Skin after a Double Pinch (1.5 h(I)/4 h(R)/1.5 h(I)/24 h(R)) 

The durations of 1.5 h(I)/4 h(R)/1.5 h(I) tested for the double pinch resulted in a Stage 2 pressure ulcer. After 24 h, spongiosis (gaps between the keratinocytes) was observed at the edge of the injury (Figure 1a–c). However, the epidermis was no longer present in the centre of the insult and was replaced by necro-inflammatory debris, which appeared as the dark purple staining of neutrophils and spent neutrophils (Figure 1c–f). Hair follicles at the edge of the insult showed no major differences but the sebaceous glands showed some loss of nuclei (Figure 1g–i). There was a lack of hair follicles in the centre of the insult and any remaining ones were surrounded by leukocytes, which appeared as small dark purple cells (Figure 1g–i). Leukocytes (dark purple cells with very little cytoplasm) were also observed in the blood vessels and in skin located away from the insult (surrounding tissue). Leukocytes were also found at the edge and the centre of the insult (Figure 1j–l). There was a lack of fibroblasts (as shown by stellate purple nuclei) in both the edge and the centre of the insult (Figure 1j–l). The panniculus carnosus (PC) also showed considerable damage at the edge of the insult and a loss of whole muscle fibres (large deep pink bundles) was observed in the central insulted region (Figure 1m–o).

### 3.3. Cytokine and MMP-9 mRNA Expression in a Single Pinch 1.5 h(I)/24 h(R) Compared to a Double Pinch 1.5 h(I)/4 h(R)/1.5 h(I)/24 h(R)

The double ischemia reperfusion cycle generated a more serious injury. To investigate the increased inflammation, mRNA of interleukin-6 (IL6) and tumour necrosis factor-alpha (TNFα) were measured and compared to the single pinch. It has been reported that matrix metalloproteinase-9 (MMP-9) protein levels are elevated in human non-healing pressure ulcers [55]. In a single pinch model, the mRNA of TNFα, IL6 and MMP-9 were not significantly increased after 24 h when compared to normal skin levels (Figure 2). However, following a double pinch insult, all three mRNA levels were significantly increased *p* < 0.05, (Figure 2).

### 3.4. Cx43asODN Treatment Prevents the Increase of Epidermal Cx43 Protein Levels after 1.5 h(I)/4 h(R) 

After 1.5 h(I)/4 h(R), the gap junction protein Cx43 becomes highly elevated in the epidermis. Cx43asODN treatment, administered subcutaneously to the insult site, allows us to prevent the elevation and investigate the role of Cx43 upregulation in sterile inflammation and ulcer progression. Immediately after 1.5 h(I), 100 μL of Cx43asODN in 30% *w*/*v* Pluronic^®^ gel or the vehicle control (VC) was injected subcutaneously under the region of the insult. After 1.5 h(I)/4 h(R), the skin was harvested, and epidermal Cx43 protein staining was measured by image analysis. Epidermal Cx43 staining increased by 4.24-fold in the insulted skin treated with the vehicle control (VC) (Figure 3b). However, the increase of epidermal Cx43 was significantly reduced (Figure 3d,e) in the Cx43asODN treated group (TR), and epidermal Cx43 levels were similar to normal levels (Figure 3e). This suggested that the treatment with Cx43asODN was able to prevent I/R-induced upregulation of epidermal Cx43. 

### 3.5. Cx43asODN Treatment Prevented Negative Epidermal Changes after 1.5 h(I)/4 h(R)

After prevention of I/R-induced Cx43 upregulation, it was of interest to observe if any negative histological changes could be prevented. In VC skin, epidermal nuclei were condensed (smaller and darker purple with reduced cytoplasm) whilst they were not in the Cx43asODN treated skin (Figure 4a). However, there was no obvious alleviation by Cx43asODN treatment observed in the dermal and hypodermal regions after 1.5 h(I)/4 h(R) (Figure 4a).

### 3.6. Cx43asODN Prevented Epidermal Spongiosis, Loss of Dermal Cells and PC Degradation after 1.5 h(I)/24 h(R)

After 1.5 h(I)/24 h(R), the VC skin showed epidermal spongiosis (clear gaps between the keratinocytes), loss of dermal fibroblasts and muscle degeneration (loss of size of muscle and loss of visible striations) (Figure 4b). In contrast, the Cx43asODN treatment resulted in no epidermal spongiosis, and more fibroblasts were observed with better morphology (Figure 4b). In addition, the PC muscle fibres were less damaged, they were larger in size and their nuclei and striations were clearly visible (Figure 4b). 

### 3.7. Cx43asODN Treatment Reduces Neutrophil Infiltration to the Dermis after 1.5 h(I)/4 h(R)

In I/R injury, sterile inflammation plays a role in removing dead cells but may also damage cells. Neutrophil infiltration was observed in the dermis of this I/R skin model, it was important to determine if Cx43asODN treatment was able to reduce neutrophil infiltration. Neutrophils were stained with a neutrophil specific Ly6g antibody, and they were observed to infiltrate the VC skin but this was a little reduced in Cx43asODN treated skin (Figure 5). 

### 3.8. HMGB1 Protein Levels Are Significantly Increased after 1.5 h(I)/4 h(R) 

Neutrophil infiltration and accumulation under the epidermis was observed after 1.5 h(I)/4 h(R) (Figure 5). It was of interest to see if there were any changes to epidermal HMGB1 protein levels as it is an early mediator of cell death and a cytokine in I/R injury. Insulted skin tissues were stained with HMGB1 (in red) to investigate any changes after 1.5 h(I)/4 h(R), the normalised HMGB1 protein signal was found to be significantly increased in the epidermis of VC skin compared to normal skin (Figure 6a,b,g). However, after1.5 h(I)/24 h(R), the epidermal HMGB1 protein signal was no longer significantly increased compared to normal levels (Figure 6d,e,h).

### 3.9. Cx43asODN Treatment Prevents the Increase of HMGB1 Protein Levels after 1.5 h(I)/4 h(R)

To investigate if treatment has any effect on HMGB1 protein levels, Cx43asODN or its vehicle control was administered immediately after 1.5 h of ischaemia. After 4 h(R), epidermal HMGB1 increased in VC skin. In comparison, HMGB1 was significantly reduced by Cx43asODN treatment (Figure 6b,c,i). After 1.5 h(I)/24 h(R), epidermal HMGB1 declined from the increased level seen at 4 h post ischaemia in VC skin (Figure 6b,e). In Cx43asODN treated skin, HMGB1 protein signals were comparable to those found in normal skin (Figure 6k). When comparing the HMGB1 signal in vehicle control and Cx43asODN treated skin, there was no significant difference (Figure 6j) 

### 3.10. Epidermal RIP3 Protein Levels Increase with I/R Progression after 1.5 h(I)/4 h(R) and 24 h(R)

RIP3 protein is responsible for necroptosis and is reported to be upregulated in I/R injuries. Skin sections were stained with RIP3 antibody. After 1.5 h(I)/4 h(R) and 24 h(R), RIP3 protein signal was increased in the VC epidermis when compared to the normal epidermis (Figure 7a–c). However, RIP3 was only significantly increased after 24 h of reperfusion but not after 1.5 h(I)/4 h(R) (Figure 7d). 

### 3.11. Cx43asODN Prevents the Increase of RIP3 Protein Levels during I/R Progression after 1.5 h(I)/4 h(R) and 24 h(R) 

To investigate if treatment with Cx43asODN has any effect on RIP3 protein levels, Cx43asODN or its vehicle control were administered immediately after 1.5 h of ischaemia. After 1.5 h(I)/4 h(R), a comparable epidermal RIP3 protein signal was observed between VC and Cx43asODN treated insulted epidermis (Figure 7f,g,k). However, after 1.5 h(I)/24 h(R), epidermal RIP3 protein signal is significantly reduced by Cx43asODN treatment when compared to the vehicle control (Figure 7i,j,l).

### 3.12. Cx43asODN Treatment Reduces PU Progression and I/R Injury

Cx43asODN treatment in the single pinch model reduced the increase in epidermal Cx43 following I/R insult, reduced epidermal nuclear condensation, neutrophil infiltration, HMGB1 and RIP3 expression. These observations indicate that Cx43asODN treatment was able to alleviate the negative effects of I/R-induced injury. Cx43asODN treatment has potential in reducing the progression of a Stage 1 PU to a more severe PU stage where the epidermis is lost. To test this, we used a double pinch model, which results in a more severe I/R injury where the epidermis is lost. This was achieved by subcutaneously administering Cx43asODN or its Cx43senseODN (Cx43sODN) control immediately after the first 1.5 h of ischaemia. Macroscopic images show that Cx43asODN treated skin produced less severe injuries compared to its control (Figure 8). The peak of progression was at days 4 and 5 and were significantly less severe in the Cx43asODN group compared to its control (Figure 8, n = 3, *p* < 0.05). By day 4, only 60% of Cx43asODN treated skin progressed into an open wound compared to 100% of Cx43senseODN control skin (Figure 8, n = 10, *p* < 0.05). By day 10, 50% of Cx43asODN treated skins were fully healed compared to only 10% of Cx43senseODN control treated skin (Figure 8). 

## 4. Discussion

### 4.1. Summary of Key Findings

Our study has shown that a single subcutaneous injection of 300 μM Cx43asODN under the insult resulted in the reduction of the pathological increase of epidermal Cx43 protein levels, neutrophil infiltration, epidermal HMGB1 and epidermal RIP3 in an experimental early PU model of a single pinch of 1.5 h of ischaemia. Further investigation of the Cx43asODN treatment found it to be able to reduce the progression of experimental PU in a I/R double pinch mouse model. 

### 4.2. Reduction of Cx43 Protein Upregulation and Histological Observations

This study is the first to examine the role of Cx43 in the context of an early experimental pressure ulcer. However, it should be noted that the levels of Cx43 varied between animals though the response to the insult was always Cx43 upregulation but to different extents. Using a single pinch model of 1.5 h of ischaemia, 4 h after Cx43asODN treatment, epidermal Cx43 upregulation was found to be significantly reduced but not completely down to base level (Figure 3). This was accompanied by the keratinocytes from the insulted skin looking normal (Figure 4). The effects of Cx43asODN treatment were monitored at 24 h and epidermal spongiosis (as shown by gaps between keratinocytes) was prevented and more dermal fibroblasts were observed compared to the vehicle control (Figure 4). These findings confirmed that preventing epidermal Cx43 upregulation was associated with preserving histological normality in the insulted skin. 

In a streptozotocin (STZ) diabetic rat wound model, an abnormal increase in epidermal Cx43 and the formation of a non-migratory bulb of keratinocytes was observed at the wound edge (day 1) [45]. By treating these wounds with Cx43asODN, abnormally high epidermal Cx43 expression at the wound edge was reduced [45]. This resulted in faster re-epithelisation in the diabetic rat wound [45]. It has been suggested that Cx43 interacts with P120ctn/Rho GTPase, which could be important for the migration of keratinocytes [56]. In human diabetic wound healing, Cx43 has been shown to persist in the wound edge in the epidermis instead of downregulating [44,57] or even increase many fold in in chronic wounds [44,57]. 

There are currently different Cx43 targeting treatments in phase II clinical trials for diabetic and non-healing wounds [58]. Rectifying the abnormal increase of epidermal Cx43 is important for the wound healing process to proceed. In this study, preventing/reducing the pathological upregulation of epidermal Cx43 by Cx43asODN in early experimental PU share similar beneficial effects with these publications [58]. 

### 4.3. Regulating Necroptosis and Preventing PU Progression

In the Cx43asODN treatment group, fewer neutrophils infiltrated the insulted region of the skin that underwent 1.5 h(I)/4 h(R) injury (Figure 5). This reduced number of neutrophils may indicate that there were fewer proinflammatory cytokines or chemokines being produced. The level of inflammation does not return to that of normal skin, but it has been dampened down which reduces the damage. It has been suggested that cell death following ischemia reperfusion is by apoptosis [59], but this does not entirely fit with the inflammatory nature of the injury, as apoptosis is generally a non-inflammatory process [59]. The possible involvement of necrosis led to the interest in HMGB1, which is a marker for necrosis [24]. Epidermal HMGB1 was found to be upregulated in skin that has been subjected to 1.5 h(I)/4 h(R) (Figure 5). This upregulation of HMGB1 was prevented in epidermal cells after Cx43asODN treatment (Figure 5). The observation that the injury was progressive pointed to some form of regulated necrosis, or necroptosis [60]. A driver protein of necroptosis, RIP3, was upregulated in the epidermis in insulted skin, which suggested that these cells were primed for necroptosis (Figure 7). The upregulation of RIP3 was prevented by Cx43asODN treatment at 4 h and 24 h (Figure 7). The upregulations of epidermal Cx43, HMGB1 and RIP3 were all prevented by Cx43asODN treatment. This suggests that Cx43asODN treatment has the potential to prevent the progression of the Stage 1 PU to a more severe PU stage where the epidermis is lost. This was confirmed when a single dose of Cx43asODN treatment was able to reduce I/R injury and limit the progression of the experimental PU (Figure 8). 

Epidermal HMGB1 was found to be 2.1× higher compared to a normal epidermis after 1.5 h(I)/4 h(R) but returned to normal after 24 h(R). HMGB1 can be upregulated in just 30 min after ischaemia in the heart, peaking at 6 h post ischaemia decreasing thereafter [31]. This is consistent with the observation that HMGB1 is elevated at 4 h of reperfusion, but not after 24 h reperfusion (Figure 5). Cx43asODN treatment prevented the increase of HMGB1 protein by a factor of 2. This could be due to reduced spreading of death signals as Cx43 upregulation was prevented by Cx43asODN treatment. There is also a possibility that Cx43 directly regulates HMGB1. Gago-Fuentes et al. showed that HMGB1 could be one of the Cx43 interacting proteins in moderate osteoarthritis (grade III) [61]. It is important to note that Cx43–HMGB1 interaction is not detected in healthy or early osteoarthritis (grade I). Reviews on Cx43 interacting proteins did not find Cx43–HMGB1 interaction to be present [62,63]. This could mean that some modifications to Cx43 such as phosphorylation or dephosphorylation may be required before it is able to interact with HMGB1 [64]. Interactions could also be prohibited due to location differences in the cell, Cx43 is located in lipid rafts while HMGB1 is normally in the nucleus. 

In contrast to HMGB1 being upregulated after 1.5 h(I)/4 h(R) and reduced at 24 h(R), epidermal RIP3 was observed to be 1.82× higher than normal after 1.5 h(I)/4 h(R) and continued to upregulate to 2.29× higher than normal after 24 h(R). This continued upregulation of RIP3 has been reported in hippocampal neuronal cultures, which have undergone 2 h of oxygen and glucose deprivation and 7 h or 24 h of reperfusion. RIP3 expression was found to be much higher at 24 h than 7 h post oxygen and glucose deprivation [65]. Here, epidermal RIP3 upregulation was not only prevented but seems to have been reduced in epidermal cells in the Cx43asODN treated group (Figure 7). There are no publications suggesting a direct regulatory link in between RIP3 and Cx43. However, RIP3 is located in the cell membrane where Cx43 is found. When RIP3 is activated, the RIP1/3 complex will recruit MLKL to the lipid rafts of cell membrane to execute necroptosis [66]. Likewise, connexins (Cx43, Cx32, Cx36, Cx46) are also targeted for lipid rafts when they are expressed [67]. 

The upregulation of epidermal Cx43 was attenuated by the treatment with Cx43asODN. It could be an indirect effect of Cx43asODN downregulating Cx43 in other cell types that resulted in reduced inflammation [68]. For example, Cx43 could be downregulated in the endothelium, resulting in reduced inflammation [69]. Cx43 could also be downregulated in dermal fibroblast to prevent the spread of cell death, which would result in a less damaged dermis [50]. A single dose of Cx43asODN administered after ischaemic insult was effective in reducing PU progression in the double pinch model. In treating a diabetic wound, a single dose was also used to stimulate re-epithelisation to kick start the healing process [45]. However, it remains unknown if administration before the ischemic insult or repeated doses would prove more effective. However, repeated administration could interfere with the normal functions of Cx43 in skin homeostasis [42,70].

The double cycle pinch model was more severe than a single pinch with significantly higher mRNA expression levels of IL-6, TNFα and MMP-9. There was variability seen in the levels of elevation and this may reflect the variations we saw in Cx43 levels between animals. IL-6 and TNFα findings were consistent with a publication by Kurose et al. that performed a genome-wide screening of rat mRNA expression of cytokines during experimental PU. They reported that GM-CSF, IL-1β, IL1Rα, IL-6, IL-10 and TNFα were increased at 12 h [71]. Human Stage 3 and 4 chronic PUs also have increased mRNA expression levels of IL-1β and TNFα [72]. MMP-2 and MMP-9 have been reported to be increased in experimental PUs in rats on day 7, 14 and 21 [73]. Increased MMP-9 activity has been correlated to the increasing severity of human chronic wounds, which is consistent with what we see here [74].

## 5. Conclusions

In conclusion, Cx43asODN treatment was found to be able to prevent the increase of epidermal Cx43 and in turn, reduce I/R injury and the progression of experimental PU. Neutrophil numbers were reduced and the upregulation of epidermal HMGB1and RIP3 was prevented. These effects translated into the successful prevention of experimental PU progression. Investigations into necroptosis suggest the possibility that Cx43 may be linked to RIP3 and HMGB1.

## Figures and Tables

**Figure 1 cells-12-02856-f001:**
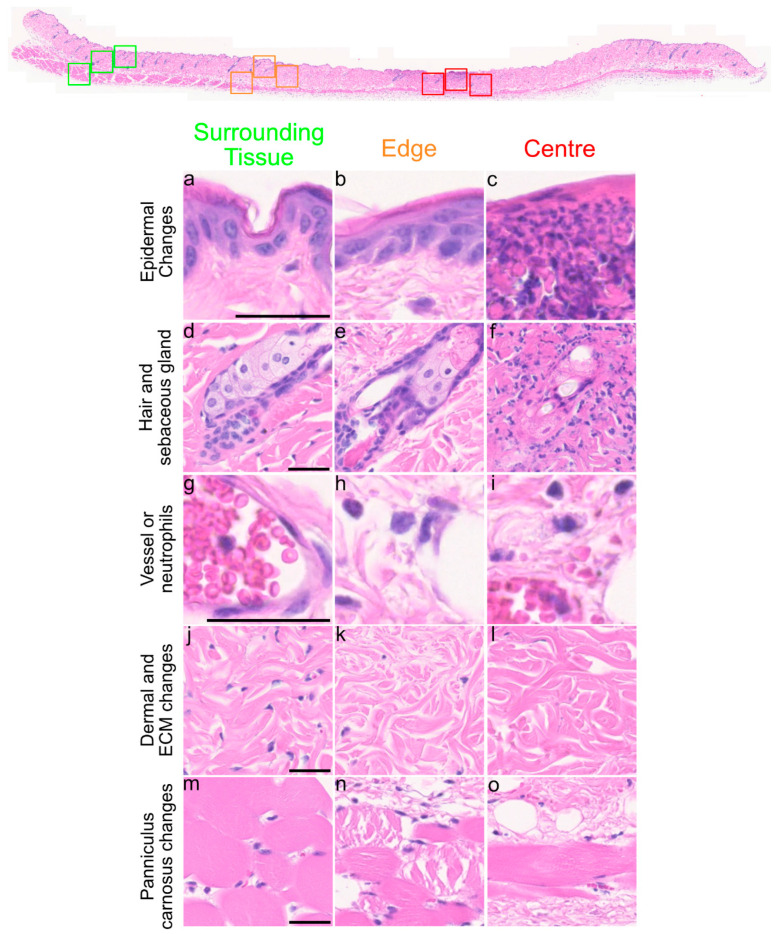
**Histological observations of skin after 1.5 h(I)/4 h(R)/1.5 h(I)/24 h(R).** Overview of skin section. Coloured squares show whether the pictures were taken from surrounding tissue (green), the edge of the injury (orange) and centre of the injury (red) and the red line above demarcates the insult area. (**a**–**c**) Spongiosis (gaps between keratinocytes) was seen at the edge of injury and necroinflammatory debris (small dark purple cell staining) was found in the centre. (**d**–**f**) Hair follicles in the centre show degeneration at 24 h(R). (**g**–**i**) Neutrophils (small dark purple cells) were found at the edge and centre of the insult. (**j**–**l**) Fibroblasts (shown by small stellate purple nuclei) were lost within the insult area and many neutrophils were present. (**m**–**o**) Damage to the panniculus carnosus (PC) was seen at the edge and centre of the insult. Scale bar = 50 μm.

**Figure 2 cells-12-02856-f002:**
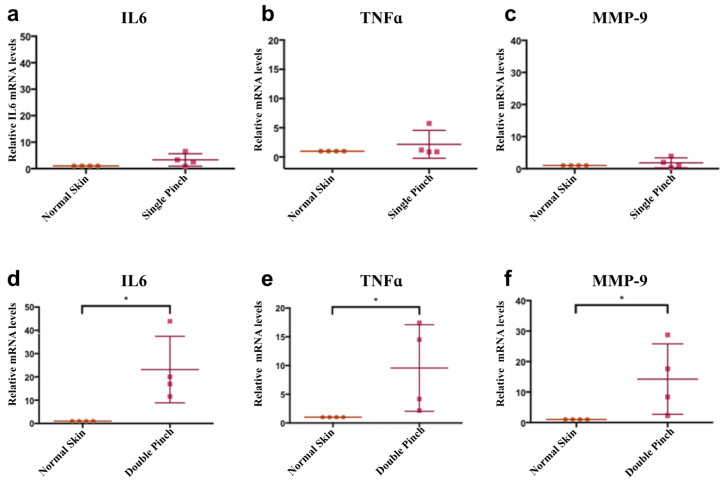
**Double pinch insult shows higher mRNA levels of IL6, TNFα and MMP-9 than a single pinch insult.** (**a**–**c**) Graphs showing relative mRNA levels of IL6, TNFα and MMP-9 of single pinch normalised to mRNA levels in normal skin. (**d**–**f**) Graphs showing mRNA levels of IL6, TNFα and MMP9 expression in a double pinch model normalised to mRNA levels in normal skin. All of them were significantly upregulated compared to normal skin levels. GAPDH mRNA levels was used as a reference gene (n = 4, *p* < 0.05, *, Wilcoxon signed rank test, mean (SD)).

**Figure 3 cells-12-02856-f003:**
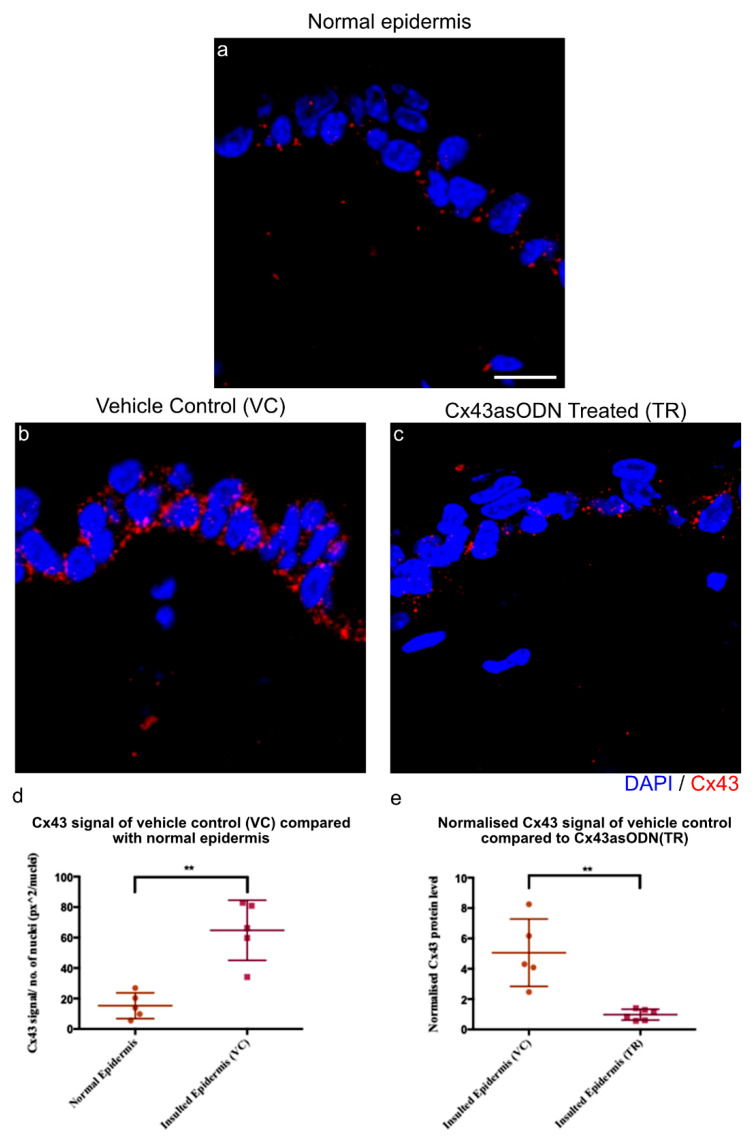
The increase of Cx43 after 1.5 h(I)/4 h(R) is prevented with Cx43asODN treatment. (**a**) Confocal images showing Cx43 (red) protein levels in the normal epidermis, (**b**) vehicle control (30% *w*/*v* Pluronic^®^) and (**c**) Cx43asODN treated insulted epidermis. (**d**) Graph showing quantified Cx43 signal of normal epidermis and VC epidermis. Cx43 was significantly increased at 1.5 h(I)/4 h(R) (n = 5, *p* < 0.01, Mann–Whiney U-test). (**e**) Graph showing quantified Cx43 signal of VC epidermis and Cx43asODN treated insulted epidermis normalised to its respective normal epidermis and compared to each other. Increase in epidermal Cx43 was significantly reduced when treated with Cx43asODN 1.5 h(I)/24 h(R) (n = 5 (VC skin), n = 6 (Cx43asODN treated insulted skin), ** *p* < 0.01, Mann–Whitney U-test). Scale bar = 10 μm.

**Figure 4 cells-12-02856-f004:**
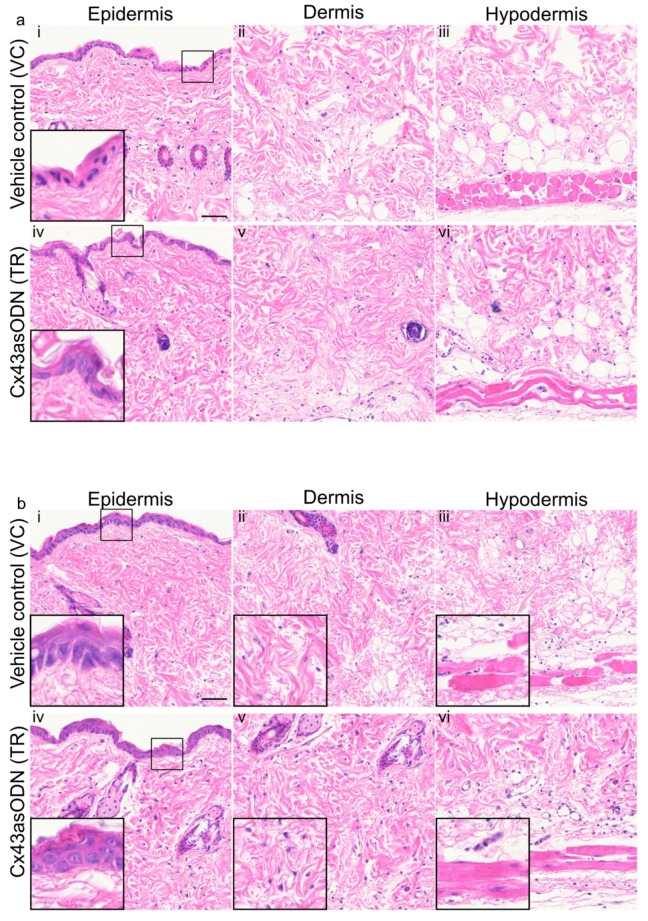
(**a**) Cx43asODN treated insulted skin after 1.5 h(I)/4 h(R) prevented epidermal nuclear condensation and (**b**) Cx43asODN treatment of insulted skin after 1.5 h(I)/24 h(R) epidermal spongiosis, loss of fibroblasts and loss of muscle nuclei. (**a**) Representative widefield images of H&E staining of the vehicle control and Cx43asODN of epidermis, dermis and hypodermis in insulted skin. (**ai**) The epidermis shows abnormal nuclei (smaller and darker purple) which is (**aiv**) prevented by Cx43asODN treatment. (**aii**,**iii**,**v**,**vi**) The dermis and hypodermis showed no obvious differences resulting from Cx43asODN treatment. n = 5 (VC skin), n = 6 (Cx43asODN treated insulted skin). Scale bar = 50 μm. (**b**) Representative images of H&E staining of the vehicle control and Cx43asODN treated insulted skin histology of the epidermis, dermis and hypodermis. (**bi**) The vehicle control epidermis shows spongiosis (gaps between keratinocytes), (**bii**) the lack of dermal cells and (**biii**) loss of muscle bulk and nuclei and striations. (**biv**) The Cx43asODN treated epidermis shows no spongiosis, (**bv**) the presence of dermal fibroblast and (**bvi**) muscle fibres with nuclei and striations are still present. n = 5 (VC skin), n = 6 (Cx43asODN treated insulted skin). Scale bar = 50 μm.

**Figure 5 cells-12-02856-f005:**
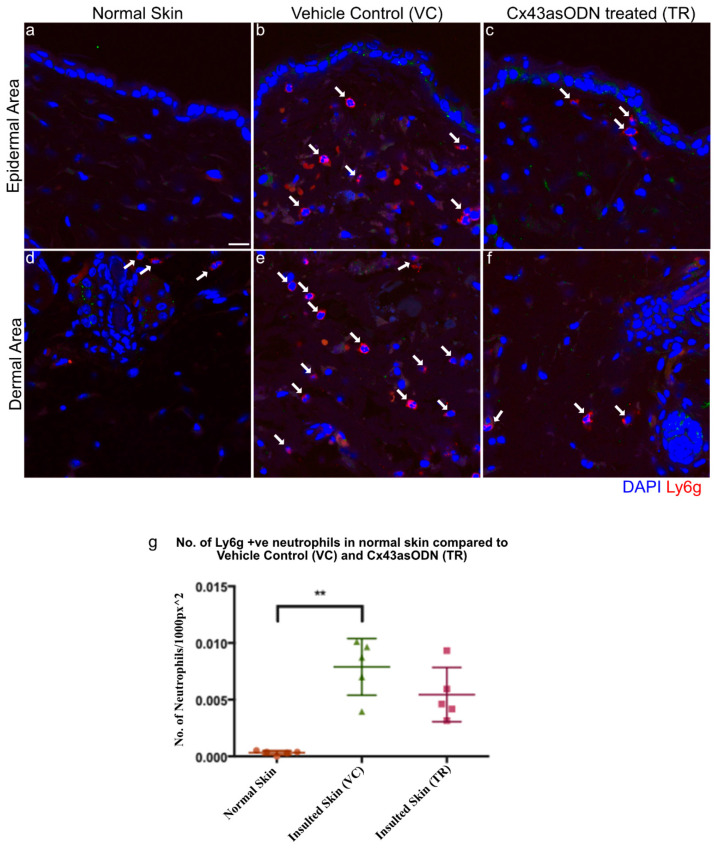
**Cx43asODN treated skin 4 h post 1.5 h of ischaemia have fewer neutrophils.** Representative confocal images of neutrophils stained in red with the Ly6g antibody, a neutrophil specific marker. White arrows point to neutrophils. (**a**,**d**) Normal skin shows almost no neutrophils and in contrast, (**b**,**e**) neutrophils were observed to have infiltrated in VC skin. (**c**,**f**) Cx43asODN treated insulted skin shows less neutrophils infiltrating the insulted region. (**g**) Graph of total neutrophils counted per 1000 px^2^ showing VC skin having a significant increase in neutrophils compared to normal skin (n = 5, *p* < 0.01, Kruskal–Wallis, Dunn’s post hoc test). Cx43asODN treated skin have a non-significant increase of neutrophils compared to normal skin. (n = 5, ** *p* > 0.01, Kruskal–Wallis, Dunn’s post hoc test). Scale bar = 25 μm.

**Figure 6 cells-12-02856-f006:**
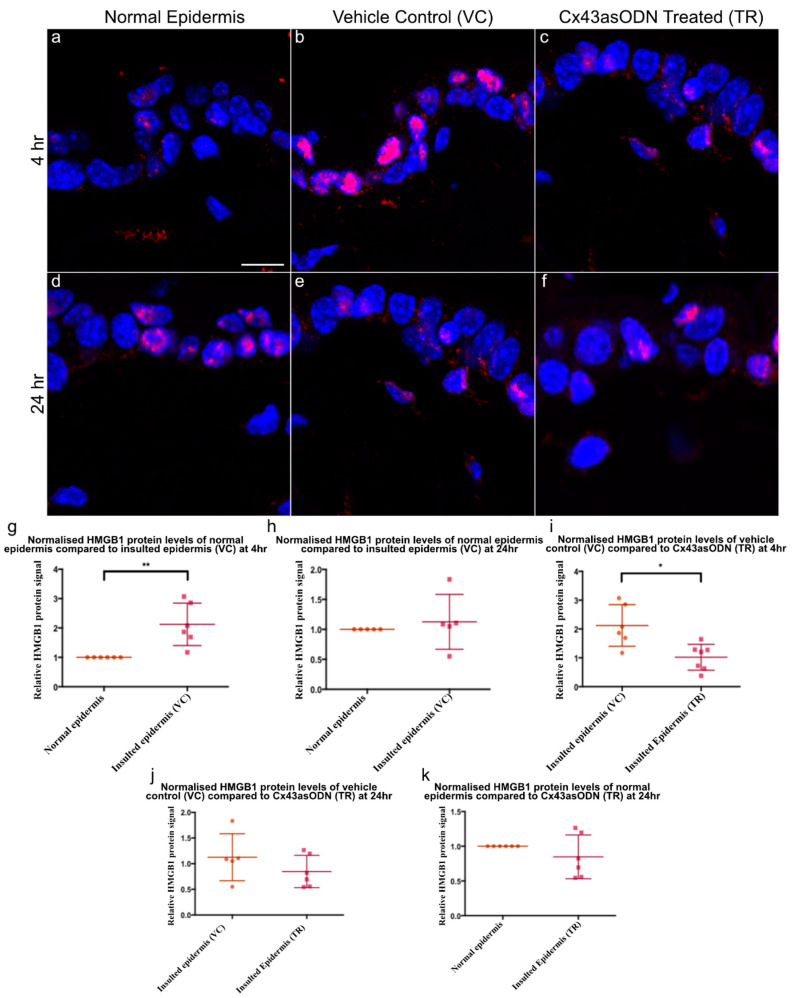
**Epidermal HMGB1 changes after 1.5 h(I)/4 h(R) and 1.5 h(I)/24 h(R).** Cx43asODN treatment prevented HMGB1 increase at 1.5 h(I)/4 h(R). Panel of representative confocal pictures showing HMGB1 stained in red in (**a**) normal skin, (**b**) VC skin (**c**) Cx43asODN (TR) skin after 1.5 h(I)/4 h(R), (**d**) normal skin and (**e**) VC skin (**f**) Cx43asODN (TR) skin after 1.5 h(I)/24 h(R). (**g**) Graphs showing quantified epidermal HMGB1 protein signal after 1.5 h(I)/4 h(R) showing significant increase (n = 6, ** *p* < 0.01, Wilcoxon signed-rank test) and (**h**) showing no significant increase 24 h later. (n = 5, * *p* > 0.05, Wilcoxon signed-rank test). (**i**) Graph of HMGB1 signal of VC and Cx43asODN treated insulted skin normalised to levels in the normal skin and compared to each other. Cx43asODN significantly prevented the increase of epidermal HMGB1 at 4 h. (n = 6 (VC skin) n = 7 (Cx43asODN treated insulted skin),* *p* < 0.05, Mann–Whitney U-test). (**j**) Graph of epidermal HMGB1 protein levels in VC and Cx43asODN treated skin normalised to HMGB1 protein levels in the normal skin and compared to each other at 24 h. (**k**) Graph of epidermal HMGB1 protein levels showing Cx43asODN treatment did not result in the upregulation of epidermal HMGB1 protein levels after 1.5 h(I)/24 h(R). (n = 5 (VC skin), n = 6 (normal skin and Cx43asODN treated insulted skin), * *p* > 0.05, Mann–Whitney U-test). Scale bar = 10 μm.

**Figure 7 cells-12-02856-f007:**
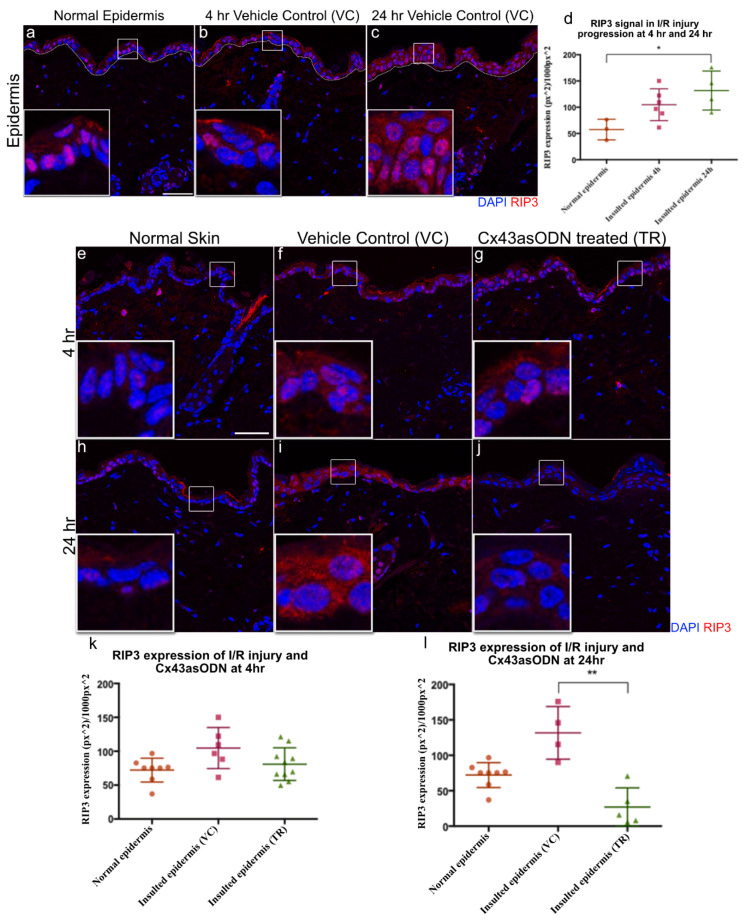
Epidermal RIP3 is significantly increased after 24 h post 1.5 h of ischaemia & Cx43asODN treatment prevented the increase of epidermal RIP3 at 24 h. Panel of confocal pictures showing RIP3 staining in red of (**a**,**e**,**h**) normal skin post ischaemia, (**b**,**f**) VC skin at 4 h post ischaemia, (**c**,**i**) VC skin at 24 h post ischaemia. (**d**) Graph showing significant increase in epidermal RIP3 protein signal after 24 h post ischaemia when compared to normal skin ((n = 3 normal epidermis, n = 6 (4 h post ischaemia vehicle control skin) n = 4 (24 h post ischaemia vehicle control skin), *p* < 0.05, Kruskal–Wallis, Dunn’s post hoc test). Panel of representative confocal images showing epidermal RIP3 staining in red of (**g**) Cx43asODN treated skin after 1.5 h(I)/4 h(R), (**j**) Cx43asODN treated skin after 1.5 h(I)/24 h(R). (**k**) Graph of quantified epidermal RIP3 protein signal after 1.5 h(I)/4 h(R) of normal skin, VC skin and Cx43asODN treated skin showing no significant increase or rescue. n = 8 (normal epidermis), n = 6 (1.5 h(I)/4 h(R) vehicle control skin), n = 9 (1.5 h(I)/4 h(R) Cx43asODN treated insulted skin),* *p* > 0.05, Kruskal–Wallis, Dunn’s post hoc test). (**l**) Graph of quantified epidermal RIP3 protein signal after 1.5 h(I)/24 h(R) of normal skin, VC skin and Cx43asODN treated skin showing a significant reduction of epidermal RIP3 protein signal (n = 8 normal epidermis, n = 4 (1.5 h(I)/24 h(R) post ischaemia vehicle control skin), n = 5 (24 h post ischaemia Cx43asODN treated insulted skin), ** *p* < 0.01, Kruskal–Wallis, Dunn’s post hoc test). Scale bar = 50 μm.

**Figure 8 cells-12-02856-f008:**
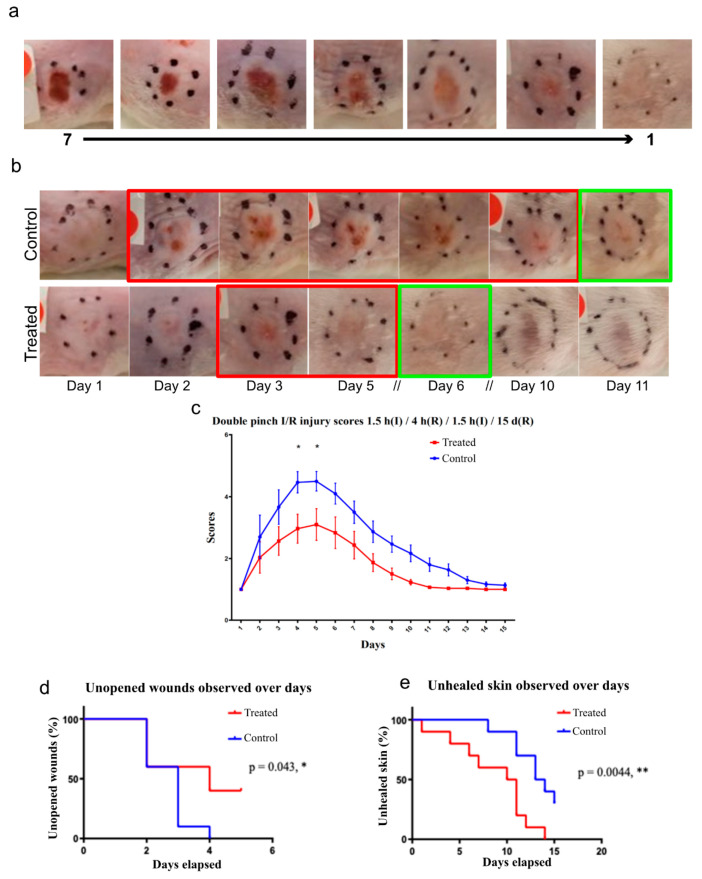
**Cx43asODN treatment reduces I/R injury progression in a double pinch model of 1.5 h(I)/4 h(R)/1.5 h(I)/15 d(R).** (**a**) Scoring system based on macroscopic appearance that includes a gradient from 1 to 7 with 7 being most severe. (**b**) Representative macroscopic picture of insulted skin that underwent the double pinch model over 11 days. Control treated skin took about 11 days to heal as opposed to Cx43asODN treated skin healed by day 6. Red boxes indicate injury phase and green box indicated that it is healed. (**c**) Pictures are scored by three independent blinded researchers and Cx43asODN treated skin is significantly less severe on days 4 and 5. (n = 3, *p* < 0.05, Two-way ANOVA, Bonferroni test) (**d**) Data was plotted as a survival curve. By day 4, significant decrease in percentage of Cx43asODN treated skin have open wounds (60%) as compared to 100% of vehicle control administered skin. (n = 10 (5 animals, 2 wounds each),* *p* < 0.05, Log-rank (Mantel-Cox) test) (**e**) Data plotted as a survival curve. By day 10, significant decrease in percentage of Cx43asODN treated mice that are fully healed (50%) as compared to just 10% of the vehicle control treated mice. (n = 10 (5 animals, 2 wounds each),** *p* < 0.01, Log-rank (Mantel-Cox) test).

## Data Availability

Can be made available by contacting the corresponding author.

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
