# Peer review of "Targeting Cx43 to Reduce the Severity of Pressure Ulcer Progression"

_cells, 2023, doi:10.3390/cells12242856_

Round 1
Reviewer 1 Report
Comments and Suggestions for Authors
The authors describe in this manuscript the involvement of connexin 43 in the healing of pressure ulcers. The inhibition of connexin 43 reduced inflammation, necroptosis, and tissue damage in mice.
The methods are overall well described. The results are precisely stated. The results are appropriately discussed.
Overall, the study would have benefitted from more detail.
Concerns
It is unclear how H&E images were analyzed. How were neutrophils, fibroblasts and necro-inflammatory debris identified?
It would have strengthened the manuscript considerably if the IF data had been confirmed with another method e.g. western blot analysis.
The terms spongiosis and abnormal nuclei need to be better explained.
To complement the reduction of the number of neutrophils with Cx43asODN a PCR panel with IL6, TNFa, MMP-9 would emphasize the effect of Cx43asODN.
Page 1 Double-check the manuscript for misspellings like targeting in the title line 2
Line31 localized damage
Comments on the Quality of English LanguagePlease double-check for misspellings. English is appropriate
Author Response
Reviewer 1
The authors describe in this manuscript the involvement of connexin 43 in the healing of pressure ulcers. The inhibition of connexin 43 reduced inflammation, necroptosis, and tissue damage in mice.
The methods are overall well described. The results are precisely stated. The results are appropriately discussed.
Overall, the study would have benefitted from more detail.
Concerns
It is unclear how H&E images were analyzed. How were neutrophils, fibroblasts and necro-inflammatory debris identified?
- We have described in the text of the results and figure legends as to how these features were identified.
It would have strengthened the manuscript considerably if the IF data had been confirmed with another method e.g. western blot analysis.
- We did not use western blot analysis because the changes in IF signals are very specific to particular tissue and cell types which required imaging and selection of regions of interest for analysis. A western blot would be unable to do this without blurring the changes.
The terms spongiosis and abnormal nuclei need to be better explained.
- This has been described in the text of the results and figure legends as to how these features were identified.
To complement the reduction of the number of neutrophils with C×43asODN a PCR panel with IL6, TNFa, MMP-9 would emphasize the effect of C×43asODN.
- This was performed and is shown in figure 2.
Page 1 Double-check the manuscript for misspellings like targeting in the title line 2
- Targeting is spelt correctly according to the Oxford English dictionary. We have been through the manuscript with a spell check. Though looking at the MDPI formatted manuscript I see it has introduced “Targetting” in the title – this is incorrect and was not how it was spelt in our submitted manuscript.
Line31 localized damage
- The submitted manuscript was spelt correctly with “localised” according to the Oxford English dictionary.
Reviewer 2 Report
Comments and Suggestions for Authors
This is an interesting paper that extends the evidence for a therapeutic benefit of Cx43 ODNs in the skin. In general, the experiments are well performed and justify the conclusions. My only significant concerns relate to the variability of some of the data.
Specific comments:
1. It is disappointing that the abstract does not include more of the data. It seems too general and vague.
2. The authors should discuss the wide variability of some of the measurements. For example, ther are big variations of the IL6, TNF, and MMP-9 levels in Fig. 2.
3. IN fig. 3: Why aren’t all the measurements and data scales comparable? Why is the normal higher in 3C than in 3D? Why is the control/insulted different between 3d and 3f?
4. Can Cx43 protein be quantified in another way (like immunoblotting)?
5. As shown in Fig. 5, ODN treated does not return the neutrophils in treated insulted skin to normal.
Author Response
Reviewer 2
This is an interesting paper that extends the evidence for a therapeutic benefit of Cx43 ODs in the skin. In general, the experiments are well performed and justify the conclusions. My only significant concerns relate to the variability of some of the data.
Specific comments:
- It is disappointing that the abstract does not include more of the data. It seems too general and vague.
- We agree. We have added many more details of the results to the abstract rather than just general features.
- The authors should discuss the wide variability of some of the measurements. For example, ther are big variations of the IL6, TF, and MMP-9 levels in Fig. 2.
- We believe that the initial variation in Cx43 levels between mice (shown in fig 3) may be responsible for the variations we see in IL6, TNF and MMP9. We have added this into the discussion.
- IN fig. 3: Why aren't all the measurements and data scales comparable? Why is the normal higher in 3C than in 3D? Why is the control/insulted different between 3d and 3f?
- There was quite a lot of variation in Cx43 levels between mice which necessitated normalising the data in 3F. We agree that 3E is confusing so we have removed this part of the figure as it is not essential.
- Can C×43 protein be quantified in another way (like immunoblotting)?
- Sorry western blotting would not work. The elevation of Cx43 levels is not global or even in distribution. We had to use immunostaining so that we could place a region of interest around the epidermis for analysis of just the epidermis. If the dermis tissue had been included in a western blot it would have masked the changes.
As shown in Fig. 5, OD treated does not return the neutrophils in treated insulted skin to normal.
- Yes the neutrophil level does not return to that of normal skin. We are just dampening down the inflammatory response not removing it all together. We have added this point into the discussion.